# Enhanced Antioxidant and Anti-Inflammatory Effects of Self-Nano and Microemulsifying Drug Delivery Systems Containing Curcumin

**DOI:** 10.3390/molecules27196652

**Published:** 2022-10-06

**Authors:** Liza Józsa, Gábor Vasvári, Dávid Sinka, Dániel Nemes, Zoltan Ujhelyi, Miklós Vecsernyés, Judit Váradi, Ferenc Fenyvesi, István Lekli, Alexandra Gyöngyösi, Ildikó Bácskay, Pálma Fehér

**Affiliations:** 1Department of Pharmaceutical Technology, Faculty of Pharmacy, University of Debrecen, Nagyerdei körút 98, H-4032 Debrecen, Hungary; 2Doctoral School of Pharmaceutical Sciences, University of Debrecen, Nagyerdei körút 98, H-4032 Debrecen, Hungary; 3Institute of Healthcare Industry, University of Debrecen, Nagyerdei körút 98, H-4032 Debrecen, Hungary; 4Department of Pharmacology, Faculty of Pharmacy, University of Debrecen, Nagyerdei körút 98, H-4032 Debrecen, Hungary

**Keywords:** curcumin, self-emulsifying systems, drug delivery, anti-inflammatory effect, antioxidant effect

## Abstract

Turmeric has been used for decades for its antioxidant and anti-inflammatory effect, which is due to an active ingredient isolated from the plant, called curcumin. However, the extremely poor water-solubility of curcumin often limits the bioavailability of the drug. The aim of our experimental work was to improve the solubility and thus bioavailability of curcumin by developing self-nano/microemulsifying drug delivery systems (SN/MEDDS). Labrasol and Cremophor RH 40 as nonionic surfactants, Transcutol P as co-surfactant and isopropyl myristate as the oily phase were used during the formulation. The average droplet size of SN/MEDDS containing curcumin was between 32 and 405 nm. It was found that the higher oil content resulted in larger particle size. The drug loading efficiency was between 93.11% and 99.12% and all formulations were thermodynamically stable. The curcumin release was studied at pH 6.8, and the release efficiency ranged between 57.3% and 80.9% after 180 min. The results of the MTT cytotoxicity assay on human keratinocyte cells (HaCaT) and colorectal adenocarcinoma cells (Caco-2) showed that the curcumin-containing preparations were non-cytotoxic at 5 *w*/*v*%. According to the results of the 2,2-diphenyl-1-picrylhydrazyl (DPPH) and superoxide dismutase (SOD) assays, SNEDDS showed significantly higher antioxidant activity. The anti-inflammatory effect of the SN/MEDDS was screened by enzyme-linked immunosorbent assay (ELISA). SNEDDS formulated with Labrasol as surfactant, reduced tumor necrosis factor-alpha (TNF-α) and interleukin-1 beta (IL-1β) levels below 60% at a concentration of 10 *w*/*w*%. Our results verified the promising use of SN/MEDDS for the delivery of curcumin. This study demonstrates that the SN/MEDDS could be promising alternatives for the formulation of poorly soluble lipophilic compounds with low bioavailability.

## 1. Introduction

*Curcuma longa* (*C. longa*, Turmeric) has been used for decades both externally and internally because of its health benefits. A polyphenolic compound called curcumin (1,7-bis(4-hydroxy-3-methoxyphenyl)-1,6-heptadiene-3,5-dione), the main active ingredient in the plant, can be isolated from the rhizome of turmeric. Based on literature data, curcumin may be promising in the treatment of various clinical conditions, and its low toxicity has also been demonstrated. Antioxidant, anti-inflammatory, neuroprotective, hypoglycemic, antitumor, hepatoprotective and cardioprotective effects have been reported in the literature [1,2]. Furthermore, several studies have already shown that the active ingredient in *C. longa* can also be used in the treatment of various dermatological diseases, for example it can be effective in the treatment of acne, atopic dermatitis, psoriasis and vitiligo [3].

Its health benefits are mainly due to its antioxidant and anti-inflammatory effects. Curcumin targets multiple signaling pathways while also showing activity at the cellular level [4]. Several studies have shown that the mechanism by which curcumin exerts its anti-inflammatory effect is to attenuate the inflammatory response of tumor necrosis factor alpha (TNF-α)-stimulated human endothelial cells by the inhibition of the TLR4 pathway and by interfering with the nuclear factor kappa B (NF-κB) signal [5,6]. Other studies suggest that the anti-inflammatory effect of curcumin is likely mediated by the inhibition of cyclooxygenase-2 (COX-2), lipoxygenase (LOX), and inducible nitric oxide synthase (iNOS) which are important enzymes in the mediation of the inflammatory processes [7].

Oxidative stress in cells caused by reactive oxygen species (ROS) may lead to metabolic dysfunctions, including loss of cell integrity and instability of enzyme function. These can eventually lead to the development of inflammatory processes. That is why antioxidants also play an important role in the treatment of inflammatory diseases [7,8,9]. The antioxidant activity of the curcumin has been reported in the literature and has been observed in several preclinical studies [10]. In most clinical studies, the effect of oral administration of curcumin at a dose of 80 to 1000 mg daily is being investigated. Tablets are commonly used as a form of curcumin administration, while in some studies curcumin was administered in the form of nanoparticles. Ghazimoradi et al. studied the use of a 1 g dose of curcumin-containing food supplement for 42 days. It was found that the treatment significantly improved the pro-oxidant-antioxidant balance in the intervention group [11].

However, the extremely poor solubility of curcumin can severely limit the bioavailability of the drug. The main problem in the formulation of dosage forms containing a natural active ingredient, is that the physicochemical properties of the biologically active component result in poor solubility and low penetration through biological membranes. As a result, the absorption of the active ingredient also deteriorates. Moreover, the bioavailability of an active ingredient may also be affected by the rate of release of the active ingredient from a given dosage form [12].

To improve the bioavailability of curcumin many drug delivery approaches have been developed including liposomes [13], nanoparticles [14], microemulsions [15], and solid dispersions [16]. However, the complicated process of complexation and encapsulation can limit their practical utilization, which is why we chose to develop self-nano and microemulsifying systems [17]. These drug delivery systems do not contain water and hence, they have improved physical and chemical stability on long-term storage [18]. Moreover, these formulations can be easily filled into capsules and have the ability to facilitate rapid absorption of the drug, which results in a quick onset of action [19]. Another advantage is that the formulation requires simple and economical manufacturing facilities, such as a simple dispenser with a magnetic stirrer [20,21].

Self-emulsifying drug delivery systems (SEDDS) are isotropic mixtures of oils, surfactants (HLB > 12), and cosolvents. They can be used to formulate various dosage forms to improve the oral or dermal absorption and increase the solubility of poorly soluble hydrophobic drugs [22]. These formulations can be classified into self-emulsifying, self-microemulsifying (SMEDDS), and self-nanoemulsifying drug delivery systems (SNEDDS) depending on the size of the spheres in the aqueous dispersion. SMEDDS are spontaneously emulsified in situ when exposed to gastrointestinal tract (GIT) fluids, forming oil-in-water microemulsions with droplet sizes of 100–250 nm. Self-nanoemulsifying drug delivery systems, in comparison, are emulsions that have a globule diameter in the nanometer range (below 100 nm) [22,23].

This study was conducted to formulate curcumin containing SNEDDSs and SMEDDSs with the help of the appropriate excipients to enhance the bioavailability of the lipophile active substance. Labrasol or Cremophor RH 40 as surfactants, Transcutol P as co-surfactant and isopropyl myristate (IPM) as the oily phase were used in our compositions. Pseudoternary phase diagrams of the formulated compositions were constructed in order to determine the ratio of the surfactant, co-surfactant and oil phase and to identify the self-nano/microemulsifying region. As SNEDDSs and SMEDDSs often proved to be thermodynamically unstable [24], the prepared formulations were subjected to heating-cooling cycles, centrifugation and freeze–thaw cycles in order to investigate the stability of them. In order to certify the safety profiles of the developed formulations, the 2-(4,5-dimethyl-2-thiazolyl)-3,5-diphenyl-2H-tetrazolium bromide (MTT) viability test on immortalized human keratinocytes cells (HaCaT) and human colorectal adenocarcinoma cells (Caco-2) was also performed. The activity of the superoxide dismutase (SOD) enzyme in HaCaT and Caco-2 cells and the free radical scavenging activity were also measured to demonstrate the beneficial antioxidant effect of the compositions. The anti-inflammatory activity of SM/NEDDSs was investigated by TNF-α and IL-1β ELISA tests.

## 2. Results

### 2.1. Solubility Studies 

The results of solubility studies of curcumin in different vehicles are presented in Figure 1. The maximum solubility of the lipophilic drug was observed in the oily phase (IPM) of the systems with a value of 256.2 ± 9.8 mg/mL. This solubility was significantly higher than that in other components studied. It can be concluded that the oily component can solubilize the curcumin in the highest amount, so it can be considered as the main excipient of the SNEDDS/SMEDDS. Among the two surfactants, Labrasol showed better solubilizing properties for curcumin (182.3 ± 8.1 mg/mL), which is probably due to the smaller hydrophilic lipophilic balance (HLB) value of this material, as Labrasol has a HLB value of 12, while Cremophor RH 40 has a value of 15 [25,26]. Comparing the surfactants (Labrasol/Cremophor RH 40) with the co-surfactant (Transcutol P), the co-surfactant has significantly higher solubility of curcumin.

### 2.2. Emulsification Efficiency of Surfactants and Co-Surfactants

The percentage transmittance of the various dispersions are presented in Figure 2. The highest transmittance value considering the surfactants alone, was observed with the use of Labrasol (88.21% ± 0.48%). The transmittance value of the dispersion formulated with the other surfactant (Cremophor RH 40) was significantly less (54.32% ± 0.24%) than that observed for Labrasol. According to the results it can also be concluded that the emulsion prepared using Labrasol as surfactant and Transcutol P as co-surfactant showed higher transmittance value then the emulsion made with Cremophor RH 40 and Transcutol P.

### 2.3. Evaluation of Self-Emulsifying Systems

Pseudoternary phase diagrams were constructed by using a conventional water titration technique to follow up the changes in the SNEDDS/SMEDDS systems.

IPM as oily phase, Labrasol or Cremophor RH 40 as surfactant, and Transcutol P as co-surfactant were used in our compositions for the base. The maximum nano/microemulsion existing zones observed are shown in Figure 3.

The size of the nano/microemulsion region in the diagrams was compared, the larger the size the greater the self-nano/microemulsification efficiency. Based on the results it can be concluded that compositions with a lower amount of the oily component (B0 and D0) had a larger nano/microemulsion existing zone, so they were more advantageous as a self-emulsifying drug delivery system. The diagrams showed that the zone of nano/microemulsion (the grey area) was the largest in case of the composition prepared with Labrasol-Transcutol P-IPM mixture at 1:1:1 ratio.

### 2.4. Droplet Size, Polydispersity Index and Zeta Potential

The diameter of the dispersed phase was investigated by a Cumulant Dynamic Light Scattering (DLS) device. Droplet size and polydispersity index (PDI) were also determined. The values obtained, i.e., the intensity (%) as a function of size (diameter, nm), are plotted in the graph below. (Figure 4) The droplet sizes of the formulations with higher amounts of surfactant and co-surfactant were in the range below 100 nm, so in these cases (formulation A0, A, C0, C) we can talk about self-nanoemulsifying drug delivery systems. Higher droplet sizes were measured for formulations with the active ingredient in every case. According to the results the particle size of the formulations containing higher amounts from the oily phase were significantly larger than the size of the compositions with less IPM. Formulations B, B0 D and D0 are in the range of 0.2–0.5 micrometers so—according to the literature—in these cases we can talk about self-microemulsifying drug delivery systems (SMEDDS) [24,27]. We found that the droplet size of the disperse phase depends on the type of the composition of the formulation.

The measured values of the droplet size, polydispersity index and the zeta potential are presented in Table 1. According to our measurements, the addition of curcumin did not significantly affect the size of the self-emulsifying system in either case.

It was also found that all SNEDDS/SMEDDS compositions were monodisperse systems with zeta potential ranging from −15.12 ± 0.44 mV to −18.12 ± 0.36 mV. As can be seen in Table 1. the droplet size increased with the addition of curcumin, however the PDI values were unchanged (<0.23), indicating a homogeneous size distribution. Based on the results obtained, the use of Labrasol as surfactant led to nano/microemulsions demonstrating lower polydispersity values compared with the formulation containing Cremophor RH 40.

The higher the zeta potential in absolute value, the greater the stability because the increased surface charge inhibits the aggregation of the particles. According to the results of the zeta potential measurements, SNEDDS were significantly more stable than SMEDDS. It was found that the addition of the active ingredient did not significantly affect the zeta potential.

### 2.5. Determination of the Drug Loading Efficiency

The entrapped curcumin content in self-nano/microemulsions was calculated from the equation described in the Section 4.6. According to our investigation, the drug loading efficiency was close to 100% in both cases as presented in Table 1. There was no significant difference between the curcumin concentrations in the SNEDDS and SMEDDS formulations. The drug loading efficiency for all SNEDDS/SMEDDS–curcumin formulations were found in the range of 93.11% ± 0.25% to 99.12% ± 0.43%, indicating uniform drug dispersion in the compositions. It was justified that there was no significant difference in drug content among the various formulation. It was observed that composition A and C have the highest curcumin content. This may be attributed to a higher concentration of oily phase (IPM) in these two SNEDDS formulations that possess higher capacity to solubilize the 10 mg dose of curcumin.

### 2.6. Thermodynamic Stability Tests

According to the results of the thermodynamic stability tests, our formulations showed no precipitation, creaming, cracking or phase-separation after the heating–cooling cycles, centrifugation and freeze–thaw cycles as shown in Table 2. By visual observation, it was found that the physical appearance of all SMEDDS and SNEDDS remained unchanged even after the tests. According to the results it can be stated that the thermodynamic stability of the formulated SNEDDS and SMEDDS were acceptable.

### 2.7. In Vitro Dissolution Study

The curcumin release from SNEDDS and SMEDDS at pH 6.8 is shown in Figure 5. According to our study, compositions with a droplet size in the nano-range (A and C) were able to provide better drug release and dissolution than SMEDDSs. It can be concluded that the selected oil, surfactants, and co-surfactant were able to provide sufficient solubility enhancement to the curcumin. From capsules filled with the curcumin extract alone, significant drug release could not be detected. In case of the compositions A and C, drug release reached more than 50% within 90 min. Moreover, in the case of formulation C, the curcumin release attained a value of more than 80% at 180 min, which was considered an excellent result. Around 20% of the drug remained unreleased after 180 min of the study in terms of formulation C, which contained Labrasol as surfactant. The dissolution profile of the curcumin from compositions B and D was very similar, which was believed to be due to the same surfactant–oil ratio of these self-microemulsifying systems.

### 2.8. In Vitro Cell Viability Assays

MTT cytotoxicity tests were carried out on HaCaT and Caco-2 cell monolayers. The cytotoxicity of 1, 5, 10 (*w*/*v*)% solutions of the formulated SNEDDS/SMEDDS and the curcumin extract was determined. The different compositions were dissolved in PBS. As can be seen in Figure 6a,b, the amount of the surfactant used in the formulation of self-emulsifying systems affected the cell viability significantly. The most significant decrease in the viability was detected in the cases of the compositions B0, B, D0 and D, where higher amounts of surfactants were used during the formulation. Treatment with formulation C0 and C, showed the highest HaCaT and Caco-2 cell viability. The measured values were higher than 70% even when cells were treated with the 10% solution. According to these results, it was found that compositions formulated with Labrasol were better tolerated by the cells. There were no significant differences between C and C0 (1%)-treated and PBS-treated (negative control) groups, while in the case of HaCaT cells, neither treatment with formulation A cause a significant decrease in the viability. Comparing the two cell lines, we found that HaCaT cells tolerate the treatment better than Caco-2 cells. According to the results, the application of curcumin did not statistically change the cell viability values in any of the cases. The treatment with the curcumin extract did not cause toxicity to keratinocytes or colon epithelial cells at either concentration, so the active substance itself can be considered safe.

### 2.9. In Vitro Antioxidant Capacity

#### 2.9.1. Superoxide Dismutase Activity

The SOD activities of the treated groups were compared with the enzyme activities of the control cells which were treated only with PBS. According to the results, SOD activity was significantly higher in the samples containing curcumin. It was also found that pretreatment was more effective in the case of the keratinocyte cells compared with the colorectal cells as shown in the Figure 7a,b. The best result was obtained with the pretreatment of composition C in the case of the HaCaT cells (0.822 U/mL), and composition D for Caco-2 cells (0.652 U/mL). In these formulations Labrasol was used as surfactant, which may have helped the active substance to enter the cells better than Cremophor RH 40. Sod activity of Caco-2 and HaCaT cells treated with curcumin extract (Cur-extr) was significantly lower than that of cells treated with curcumin-containing SNEDDSs or SMEDDSs.

#### 2.9.2. DPPH Radical Scavenging Activity

The percentage of antioxidant activity (AA%) of the formulations with or without curcumin was determined based on the DPPH activity assay (Figure 8). In comparison, the antioxidant activity of the curcumin-containing compositions was significantly higher than the activity of the same compositions but without curcumin. SNEDDSs and SMEDDSs without the active ingredient did not show significant radical scavenging activity.

According to the results, composition A and C, which were self-nanoemulsifying drug delivery systems, showed more effective antioxidant activity in the DPPH assay. Moreover, for composition C, antioxidant activity above 50% was detected. The results indicated that curcumin had more antioxidant activity when it was used in a SNEDDS composition. In this study it was found that the curcumin as an extract had significantly lower radical scavenging activity than curcumin incorporated to SNEDDS or SMEDDS.

### 2.10. In Vitro Anti-Inflammatory Effect

The anti-inflammatory effect of the SNEDDSs and SMEDDSs with or without curcumin and the effect of the curcumin extract were studied by ELISA tests on Caco-2 and HaCaT cell lines. In the study, PBS was selected as the negative control. It was taken as 100% during the evaluation and the values were compared and expressed as the percentage of the control. As it was presented in Figure 9 and Figure 10, pretreatment with the compositions without the active ingredient did not significantly influence the level of IL-1β or TNF-α in the cells. According to the results of the study, in the case of Caco-2 cell line, preparations containing curcumin have been shown to be more effective in terms of anti-inflammatory effect, in comparison with the HaCaT cells. Composition C, which was formulated with Labrasol as surfactant, showed the most potent anti-inflammatory effect at every concentration, both TNF-α and IL-1β levels decreased the most as a result of treatment with this formulation. There were significant differences between the cytokine level of the cells which were treated with the curcumin containing SNEDDS/SMEDDS and with PBS.

Based on the results, it can be concluded that the degree of anti-inflammatory effect increased in direct proportion to the concentration used, since the higher concentration of the curcumin containing systems resulted in a higher anti-inflammatory effect in both HaCaT and Caco-2 cells. According to the results, treatment with the curcumin extract resulted in a smaller decrease in the TNF- α and IL-1β levels of both cell lines, than treatment with the curcumin containing SNEDDSs or SMEDDSs.

## 3. Discussion

The present study describes the development of novel curcumin-containing drug delivery systems and the investigation of their antioxidant and anti-inflammatory potential. As it has been described in the literature, the formulation of self-nano/microemulsifying drug delivery systems is an effective method for enhancing the *per os* and the *percutaneous* bioavailability of various poorly soluble drugs [28]. The primary objective of our experimental work was to formulate self-nano/microemulsifying drug delivery systems (SN/MEDDS) with the help of surfactants (Labrasol, Cremophor RH 40), co-surfactant (Transcutol P) and isopropyl myristate in order to enhance the solubility and the bioavailability of curcumin.

By the construction of the pseudoternary phase diagrams, the quantities of oil phase, surfactant and co-surfactant in appropriate portions was selected. As it can be seen on Figure 3, the micro/nanoemulsion existing zones were determined. It was found that the most dominant factor which influenced the area of the nano/microemulsion existing zone was the amount of the oily phase. According to the results the nano/microemulsion existing zone was the largest in case of the composition prepared with Labrasol-Transcutol P-IPM mixture in the same proportions. These results are correlated with the findings of similar studies, as Chaudhary et al. also found that with increased concentration of surfactant the region was increased [29].

Droplet size is an important parameter of self-emulsifying systems, since it can influence drug release, absorption and kinetic stability. Parmar et al. stated that increasing the proportion of oil component and decreasing the amount of the surfactant results in increasing the droplet size [30]. Our DLS measurement verified that the higher oil content resulted in larger particle size. Although the droplet size increased with the incorporation of the active ingredient, the PDI values remained unchanged, indicating a homogeneous size distribution. In self-emulsifying drug delivery systems, negative charge is common and it is accepted in the literature that a system is stable if its zeta potential is higher than ±30 mV [31]. Our formulations were characterized by slightly negative zeta potential, between −15.12 and −18.12 mV, which was due to the use of nonionic surfactants [32]. However, during the thermodynamic stability testing, no precipitation, creaming, cracking or phase-separation was observed, and these results overall suggest acceptable stability. The drug loading efficiency for formulations was found in the range of 93.11% to 99.12%. It was also observed that compositions with a higher concentration of oily phase (IPM) possess a higher capacity to solubilize curcumin.

Results of the in vitro drug release study showed that the dissolution rate of curcumin was greatly enhanced by using self-nano or microemulsifying systems. It was found that in case of the SNEDDSs the amount of drug release was higher compared with SMEDDSs; it reached more than 50% within 90 min. This improvement in the in vitro release rate of curcumin could be attributed to the spontaneous formation of nanoemulsion during the dissolution process, as it was also described by Nasr et al. This greater availability of dissolved curcumin from the composition could lead to higher bioavailability and absorption [33,34,35].

There are some criteria for the components of SNEDDS and SMEDDS regarding safety (biocompatibility), because the surfactants can be toxic at a certain concentration [36]. To test the biocompatibility of curcumin-containing formulations, in vitro cell viability studies were performed on CaCo-2 and HaCaT cells. The MTT assay is a well-established cytotoxicity test to demonstrate the effect of SNEDDS and SMEDDS on cell viability for both oral and topical administration [37,38]. Beloqui at al. described that the IC50 value (measured in terms of mg/mL curcumin) of curcumin loaded SNEDDS which contains Labrasol, Labrafil and Cremophor EL as surfactants was 0.66 mg/mL in case of Caco-2 cells [39]. Zhao et al. stated that treatment with low concentrations of curcumin (2.5 or 5 μM) effectively increased the viability and survival of HaCaT cells against inorganic arsenite-induced cytotoxicity as assessed by the MTT assay [40]. In our study the cytotoxicity of 1, 5, 10 *w*/*v*% solutions of the formulated SNEDDS and SMEDDS was determined. It was found that compositions formulated with Labrasol (C and D) were better tolerated by both cell lines. In the case of formulation C the measured cell viabilities were higher than 70%, even when cells were treated with the 10 *w*/*v*% solution, which complies with ISO 10993-5 recommendations.

Antioxidant assays were also performed with the formulated drug delivery systems. One of the main antioxidant enzymes is SOD, which can neutralize free radicals generated during oxidative stress, thus preventing cell damage [41]. SOD activity of HaCaT and Caco-2 cells was monitored after the treatment with the different SMEDDS/SNEDDS compositions. Treatment with Labrasol containing compositions (C and D) caused a significantly higher SOD level. Our results demonstrate that the effectiveness of the treatment was better in the case of HaCaT cells. This is probably because keratinocytes were able to absorb the active substance to a greater extent. In the case of Caco-2 cells, the highest SOD activity (0.652 U/mL) occurred during treatment with formulation D, which also contained Labrasol as a surfactant. This may be related to the solubility of curcumin, as it is more soluble in Labrasol than in Cremophor according to our solubility study. 

The non-enzymatic radical scavenging activity of the curcumin containing systems was also measured by the DPPH free-radical scavenging assay. Cui et al. stated that the curcumin loaded by lysozyme nanoparticle possessed higher free radical scavenging activity in the DPPH test than that of free curcumin. They stated that the DPPH radical scavenging activity of incorporated curcumin was 71.9% [42]. Iurciuc et al. also described that immobilized curcumin retained its antioxidant properties, moreover, the added materials had a protective role [43]. We found that the radical scavenging activity of the 5 *w*/*v*% alcoholic solution of composition C was the highest at 52.66%. Our results showed that curcumin had higher radical scavenging activity when it was incorporated to a SNEDDS.

According to the scientific literature curcumin possesses significant anti-inflammatory effect. In their recent article, Sadeghi et al. stated that the consumption of the curcumin supplement (1500 mg/day) was associated with significant improvement of the clinical outcomes in ulcerative colitis. Another study described that microencapsulated curcumin had a more promising effect in the treatment of arthritis in rats compared with the basic curcumin extract or micellar curcumin. Microencapsulated curcumin reduced the levels of immune cells (neutrophils and leukocytes), as well as pro-inflammatory cytokines (TNF-α, IL-1, and IL-6), while other formulations of curcumin had lower or no effect on arthritis progression [44]. In our study the formulated SMEDDSs and SNEDDSs were tested on previously inflamed HaCaT and Caco-2 cells. The results show that premedication with the drug delivery systems in a concentration of 5 *w*/*v*% significantly reduced both IL-1β and TNF-α levels. According to our study, in the case of the Caco-2 cell line, formulations were more effective in terms of anti-inflammatory effect, in comparison with the HaCaT cells. In a recent comparative study, it was demonstrated that curcumin containing SNEDDS and nanostructured lipid carriers significantly reduced TNF-α secretion in Caco-2 cells by LPS-activated macrophages, however they found that in vivo, only curcumin in nanostructured lipid carriers was able to significantly decrease neutrophil infiltration and TNF-α secretion [39].

Our work highlights the importance of using nano and microemulsifying drug delivery systems for lipophilic agents. As it is also described in the literature, the anti-inflammatory and antioxidant effect of the curcumin depends on the form of its delivery. It can be concluded that the incorporation of curcumin into SNEDDS or SMEDDS is a promising approach to overcome solubility and bioavailability barriers. Future in vivo and clinical studies could prove the relevance of these drug delivery systems and result in a clinically acceptable therapeutic option of many inflammatory diseases.

## 4. Materials and Methods

### 4.1. Materials

Curcumin (CAS number: 458-37-7) was obtained from Sigma-Aldrich Buchs (St. Gallen, Switzerland). Labrasol [Caprylocaproyl polyoxyl-8 glyceride, Chemical Abstracts Service (CAS number: 85536-07-8)] and Cremophor RH 40 [Polyoxyl 40 hydrogenated castor oil (CAS number: 61788-85-0)] used in the formulation of SNEDDS and SMEDDS were purchased from Gattefossé (Lyon, France) and BASF Company (Ludwigshafen, Germany), respectively. Isopropyl myristate [propan-2-yl tetradecanoate (CAS Number: 110-27-0)] was purchased from Hungaropharma Ltd., (Budapest, Hungary).

The MTT [2-(4,5-dimethyl-2-thiazolyl)-3,5-diphenyl-2H-tetrazolium bromide)] dye, Dulbecco’s Modified Eagle’s Medium (DMEM), phosphate buffered saline (PBS), trypsin from porcine, ethylene-diamine-tetra-acetic acid (EDTA), heat-inactivated fetal bovine serum (FBS), L-glutamine, 2,2-diphenyl-1-picrylhydrazyl (DPPH), absolute ethanol and (±)-6-Hydroxy-2,5,7,8-tetramethylchromane-2-carboxylic acid (Trolox) (CAS Number: 53188-07-1), TNF-α and human IL-1β ELISA Assay Kits were purchased from Sigma-Aldrich (Budapest, Hungary). Nonessential amino acid solution and penicillin–streptomycin mix, GlutaMax™ supplement, 96-well plates, and cell culture flasks were obtained from Thermo-Fisher (Darmstadt, Germany, CAS number: 156499). HaCaT (human keratinocyte cells) and Caco-2 (human colon adenocarcinoma) cell lines were obtained from Cell Lines Service (CLS, Heidelberg, Germany).

### 4.2. Solubility Studies

The aim of the solubility studies were the identification of suitable SNEDDS and SMEDDS components with good solubilizing capacity for curcumin. An excess amount of curcumin (1 g) was added to 2 mL of surfactants (Labrasol and Cremophor RH 40) and co-surfactant (Transcutol P) and to the oil component (IPM) of the SNEDDS/SMEDDS and mixed by vortexing. The mixture was kept at room temperature (23.5 °C ± 0.2 °C) for 48 h to achieve equilibrium and saturation solubility tests were performed. The equilibrated samples were centrifuged at 1200 rpm for 10 min to extract the insoluble curcumin. An aliquot of the supernatants was filtered through a membrane filter (0.45 µm) and diluted with absolute ethanol. The amount of the dissolved curcumin was quantified spectrophotometrically at 425 nm. All measurements were done in triplicate and the solubility was expressed as the mean value (mg/mL) ± SD [33].

### 4.3. Emulsification Efficiency of the Surfactants and Co-Surfactant

The surfactants and co-surfactant were tested for their efficiency in emulsifying the oil phase (IPM). For the determination of the emulsification ability of Labrasol and Cremophor RH 40, 500 μL of the selected surfactant was added to 500 μL of the IPM. Then, the mixture was vortexed and heated at 60 °C for 2 min for homogenization. A further 100 μL of mixture was diluted up to 50 mL with distilled water. The formed emulsions were allowed to stand for 2 h and their transmittance was measured by using a UV–VIS spectrophotometer at 650 nm. The percentage transmittance was calculated for each emulsion in triplicate and the average values ± SD were calculated [33].

In order to investigate the emulsification ability of the co-surfactant (Transcutol P), 500 μL of it was added to 500 μL of the selected surfactant (Labrasol/Cremophor RH 40) then 1 mL of the IPM was added to the mixture. The emulsions were also allowed to stand for 2 h and their transmittance was measured as described above.

### 4.4. Formulation and Investigation of Self-Emulsifying Systems

The poor solubility of curcumin, as described in the introduction, made it difficult for us to formulate dosage forms, so the solubility and thus the bioavailability was increased by using self-emulsifying systems. Self-nano/microemulsifying systems have been prepared by water and oil (isopropyl myristate (IPM)) dilution with the use of different surfactants (Cremophor RH 40 and Labrasol) and co-surfactant (Transcutol P) in different ratios. Labrasol or Cremophor RH 40 and Transcutol P in the appropriate amounts and IPM were mixed at 37 °C by a Schott Tritronic dispenser (SI Analytical, Mainz, Germany) combined with a Radelkis magnetic stirrer (Radelkis, Budapest, Hungary, version number: OP-912). The given concentration of curcumin was dissolved in the mixture of the selected surfactant and co-surfactant at 24.5 °C by permanent agitation. The final compositions of the formulated self-nano/microemulsifying systems were presented in Table 3. Altogether eight systems were developed with or without curcumin.

To examine possible phase separation, the formulated self-emulsifying systems were equilibrated for 24 h. An Erweka DT800 rotating paddle apparatus (Erweka Gmbh, Heusenstamm, Germany) was used to investigate the efficiency of self-emulsification of the compositions. Half a gram of the mixtures was added to 100 mL of distilled water with gentle stirring at 60 rpm, at 37 °C. The process of self-emulsification was visually monitored, while the rate of emulsification and the appearance of the emulsions obtained were examined.

For each system pseudoternary phase diagrams were constructed in order to determine the ratio of the oil phase, surfactant and co-surfactant and to identify the self-nano/microemulsifying region and follow up the changes. Each side of the triangle indicated the percentage of the components (water/IPM/surfactant and co-surfactant) in the mixture, which were increased in a clockwise direction in the diagram. Each corner of the diagram represents 100% of the given component. By varying percentage of oil, surfactant and co-surfactant, twelve formulations were prepared for the pseudoternary phase diagrams for each system without curcumin. With the help of the constructed diagrams, the extent and the nature of the nano/microemulsion region were determined. Only clear or slight bluish dispersions were considered in the nano/microemulsion region of the diagram [45,46].

### 4.5. Determination of Droplet Size, Polydispersity Index and Zeta Potential

A dynamic light scattering (DLS) device (Zetasizer Nano S, Malvern, UK, Worcestershire, serial number: MAL1226409) was used to determine the droplet size of the dispersed phase. Half a gram of the self-emulsifying systems were dissolved in 100 mL distilled water and then the sample was exposed to a monochrome light wave. When the monochrome light beam met a solution containing macromolecules, the light was scattered in all directions depending on the size and shape of the macromolecules [47]. The particle size distribution was also described with the help of the polydispersity index (PDI).

The electrostatic potential (zeta potential) of the double layer surrounding the droplets was also measured with Zetasizer Nano S equipment (Malvern, UK, Worcestershire, serial number: MAL1226409). With the investigation of the zeta potential, the stability of the self-emulsifying systems were determined. The samples were freshly diluted with 100 mL distilled water and analyzed in triplicate in all measurements.

### 4.6. Drug Loading Efficiency

In order to investigate the entrapment of the curcumin 100-100 mg from the formulated SNEDDS/SMEDDS were diluted with 100 mL absolute ethanol. The extraction of the drug from SNEDDS/SMEDDS was carried out by centrifugation at 10,000 rpm for 25 min. The supernatant was diluted with absolute ethanol (3 times) and the curcumin content was analyzed spectroscopically. The absorbance was measured at 425 nm by UV spectrophotometer (Shimadzu, Tokyo, Japan, serial number: A124256) [48]. The drug loading efficiency (%) was calculated by the following formula (Equation (1)):(1)$$\mathrm{Drug}\;\mathrm{loading}\;\mathrm{efficiency}\;\left(\%\right)=\frac{\mathrm{Amount}\;\mathrm{of}\;\mathrm{curcumin}\;\mathrm{measured}\;\mathrm{in}\;100\;\mathrm{mg}\;\mathrm{SNEDDS}/\mathrm{SMEDDS}}{\mathrm{Amount}\;\mathrm{of}\;\mathrm{curcumin}\;\mathrm{added}}$$

### 4.7. Thermodynamic Stability Studies

The prepared SNEDDS/SMEDDS were subjected to heating–cooling cycles, centrifugation and freeze–thaw cycles in order to investigate the thermodynamic stability of them. In heating–cooling cycles the SNEDDS and the SMEDDS were subjected to six cycles between 4 °C and 45 °C with storage at each temperature for 48 h. The formulations were centrifuged at 3500 rpm for 30 min and checked for phase separation, creaming, or cracking. Finally, the samples were stored at alternating temperatures of −21 °C and 25 °C, with the storage of 72 h at each temperature, for three cycles. The physical appearances of the different formulations were visually observed at the end of each stage [33,49,50].

### 4.8. In Vitro Dissolution Study

The in vitro drug release of curcumin from the formulated SNEDDS/SMEDDS and from the curcumin extract was performed using USP dissolution apparatus (Erweka, DT 800, Erweka Gmbh, Heusenstamm, Germany). The dissolution medium consisted of 900 mL of freshly prepared simulated intestinal fluid (SIF) without pancreatin (pH 6.8) maintained at 37 °C, the paddle speed was set at 100 rpm. From the different SNEDDS/SMEDDS–curcumin formulations, an amount equivalent to 20 mg of the curcumin was filled in a hydroxypropyl methylcellulose (HPMC) (Capsugel, Inc., Morristown, NJ, USA) capsule size “0”. Aliquots (5 mL) from the dissolution medium were withdrawn at regular time intervals (5, 15, 30, 60, 90, 120 and 180 min) using a calibrated disposable syringe, while the volume of the dissolution medium was kept at 900 mL by adding fresh medium [33,51]. The samples were filtered through a membrane filter of 0.45 μm pore size and the curcumin concentration was monitored via a spectrophotometric method at 425 nm using UV spectrophotometer (Shimadzu, Tokyo, Japan, serial number: A124256). Finally, the cumulative amount of curcumin released was determined. All measurements were done in triplicate.

### 4.9. Cell Culture Maintenance

Immortalized human keratinocyte cells (HaCaT) and human colorectal adenocarcinoma cells (Caco-2) were applied in the cell viability, antioxidant and anti-inflammatory assays. Cells were grown in a plastic cell culture flasks (Nunc™ EasyFlask™, Thermo-Fisher, Darmstadt, Germany, CAS number: 156499) in Dulbecco’s Modified Eagle’s Medium, supplemented with 10 *v*/*v*% heat-inactivated fetal bovine serum (FBS), 1 *v*/*v*% non-essential amino acids solution, 4 mmol/L L-glutamine, 100 IU/mL penicillin, and 100 μg/mL streptomycin at 37 °C in an atmosphere of 5% CO_2_. The culture medium was changed twice a week. The cells were routinely maintained by regular passaging. The HaCaT and Caco-2 cells used for the experiments were between passage numbers 20 and 40 [52].

### 4.10. In Vitro Cell Viability Assay

For the cell viability assay, different concentrations of solutions (1, 5 and 10 *w*/*v*%) were prepared from SNEDDS and SMEDDS using sterile phosphate buffered saline (PBS) solution. In order to compare the self-emulsifying systems with a simple curcumin extract (Cur-extr.) a concentration of 5 *w*/*v*% curcumin solution was formulated by the dissolution of 200 mg curcumin in 1 mL IPM and diluted it with PBS.

The cytotoxic effects of the formulations were examined by the MTT assay following Mosmann [53]. Caco-2 and HaCaT cells were seeded into 96-wells plates (VWR International Inc., Debrecen, Hungary) at a density of 10^4^ cells/well. After 5 days, the medium was removed, and the cells were treated with 100 µL of the test solutions for 2 h at 37 °C. After the end of the incubation period the test solutions were removed and a 0.5 mg/mL MTT solution (dissolved in PBS) was added to each well. The plate was incubated for 3 h at 37 °C. After the incubation time, the MTT dye was completely removed and 0.1 mL of an isopropanol-1 M hydrochloride acid (25:1) solution was added to each well solubilizing the created formazan crystals. The absorbance of each well was measured at 565 nm against a 690 nm reference by a Thermo-Fisher Multiskan Go (Thermo-Fisher, Waltham, MA, USA, Cat. No. N10588) microplate reader. As negative and positive controls PBS and Triton X 100 (10% *w*/*v*) solutions were used, respectively. Cell viability was expressed as a percentage of the cell viability of the untreated control cells, which were incubated with PBS for 2 h.

### 4.11. In Vitro Antioxidant Assay

#### 4.11.1. Determination of Superoxide Dismutase Enzyme Activity on Caco-2 and HaCaT Cells

For the investigation of the SOD activity HaCaT and Caco-2 cells were placed in a 12-well plate (10^5^ cells/well). As a positive control, cells were treated with (±)-6-hydroxy-2,5,7,8-tetramethylchroman-2-carboxylic acid (Trolox), a water-soluble derivative of vitamin E, which was dissolved in PBS immediately before use (10 µM). PBS was used as a negative control for the experiment. The formulated SNEDDS and SMEDDS were investigated in a concentration of 5 *w*/*v*%, diluted with PBS. Curcumin extract at a concentration of 5 *w*/*v*% was also examined.

During the treatment, the medium was removed from the cells, 200 µL of the test solution or the control sample were added, and the cells were incubated with the samples for another 20 min in a CO_2_ incubator at 37 °C. After removal of the samples, the cells were washed twice with PBS.

Cells were harvested and centrifuged at 2500 rpm for 10 min at 4 °C, then the cell pellet was homogenized with 20 mM HEPES buffer (1 mM EGTA, 210 mM mannitol, and 70 mM sucrose, pH 7.2) and centrifuged again at 3500 rpm for 5 min at 4 °C. The SOD activity of the supernatant was determined using a Cayman kit (Cayman Chemical, Ann Arbor, MI, USA, version number: 706002). The molecule used for detection was tetrazolium salt. The reaction medium added to the cells was composed of 50 mM Tris-HCl, pH 8.0, containing 0.1 mM diethylenetriaminepentaacetic acid (DTPA) and 0.1 mM hypoxanthine and 0.1 mM tetrazolium salt. The sample supernatant (10 µL) was incubated at 25 °C (2 min) and the reaction was initiated by the addition of xanthine oxidase (20 µL). Plates were incubated for 30 min on a shaker at room temperature and absorbance was read at 450 nm with a Multiskan Go microplate reader (Thermo-Fisher, Waltham, MA, USA, Cat. No. N10588). SOD activity was expressed as U/mL, which corresponds to the amount of SOD that inhibits 50% of the reduction of the tetrazolium salt.

#### 4.11.2. Determination of Antioxidant Capacity Based on 2,2-Diphenyl-1-picrylhydrazyl (DPPH) Radical Scavenging

The radical scavenging activity of the diluted samples (5 *w*/*v*%, diluted with absolute ethanol) was determined using the DPPH (M = 394.33 g/mol) free radical [54]. Two mL of DPPH radical solution (0.06 mM) in absolute ethanol was added to 900 µL of absolute ethanol. Then 100 µL of sample was added to the mixture and allowed to react for 30 min at room temperature. When DPPH accepted the hydrogen radical the reaction resulted in a color change from deep purple to light yellow [55]. Quantitative measurement of the remaining DPPH was carried out using a photometric determination (Shimadzu Spectrophotometer, Tokyo, Japan, serial number: A124256) at 517 nm [56], where absolute ethanol was used as background. As negative control, 2.0 mL of DPPH solution (0.06 mM) diluted with 1.0 mL absolute ethanol was applied, as positive control Trolox dissolved in PBS was used at a concentration of 10.0 µM. The antioxidant activity percentage (AA% = antioxidant activity) was determined according to Mensor et al. [57] (Equation (2)):AA% = 100 − {[(Abs_sample_ − Abs_blank_) × 100]/Abs_control_} (2)

### 4.12. Examination of In Vitro Anti-Inflammatory Effect

To investigate the anti-inflammatory effect of the curcumin containing SNEDDS and SMEDDS, ELISA tests were performed on HaCaT and Caco-2 cell lines. Cells were seeded on 96-well plates at a density of 10^4^ cells/well. When the cells fully grow over the wells’ membrane, culture media was removed. The cells were incubated with the samples for 1 h. Samples were prepared by dissolving curcumin containing extract (Cur-extr), SNEDDS and SMEDDS in PBS at different concentrations (2 *w*/*w*%, 5 *w*/*w*%, 10 *w*/*w*%). After samples were removed, 50 μL of IL-6 (30 ng/mL) was added to the cells and incubated overnight in order to induce inflammation. The next day the supernatant was removed, and human IL-1β (Sigma—RAB0273) and TNF-α ELISA Kits (Sigma—RAB0476) were used according to the manufacturer’s instructions.

### 4.13. Statistical Analysis

Data were handled and analyzed using Microsoft Excel 2016 (version number: 16.0.10827.20118) and GraphPad Prism (version 6; GraphPad Software, San Diego, CA, USA) and presented as means ± S.D. Comparison of multiple groups was performed with one-way ANOVA followed by either Dunnett’s multiple comparison test or Tukey’s multiple comparison test, depending upon whether the groups were compared to a given control group or to each other [52,58] Significant differences on the figures are indicated with asterisks. Results were regarded as statistically significant at *p* < 0.05.

## 5. Conclusions

Hereby, we report the first formulation and investigation of SN/MEDDS prepared from curcumin, Labrasol/Cremophor RH 40, Transcutol P and isopropyl myristate. According to the experiments, our formulations had high thermodynamic stability, and drug loading efficiency. The result of the in vitro dissolution test revealed relatively high curcumin release (>50%) in the simulated intestinal fluid (pH 6.8). All preparations greatly enhanced the anti-inflammatory effect of curcumin on Caco-2 and HaCaT cells as well as its antioxidant capacity while maintaining acceptable cell viability. Thus, the formulation of curcumin into SN/MEDSS could be a promising approach for overcoming the solubility and bioavailability problems related to this highly lipophilic drug. Further in vivo experiments are needed to verify improved oral bioavailability, however these in vitro results clearly indicate that SN/MEDDS formulations greatly enhance the anti-inflammatory and antioxidant capabilities of curcumin compared to other possible delivery options.

## Figures and Tables

**Figure 1 molecules-27-06652-f001:**
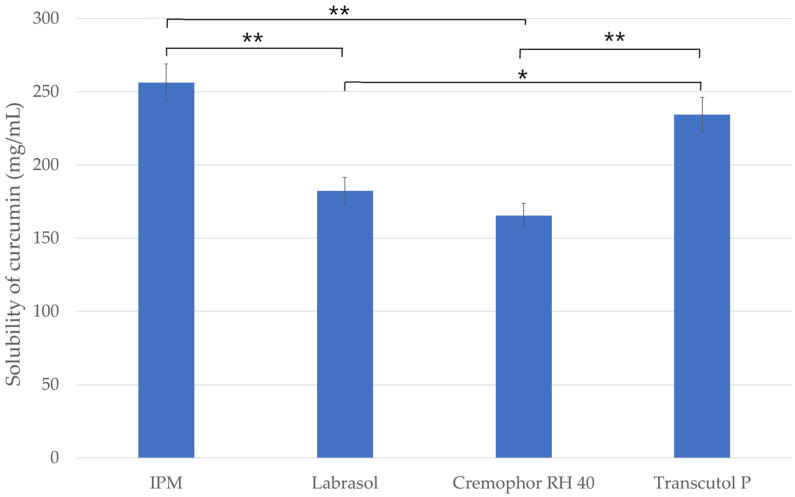
Screening of oil, surfactants and co-surfactant based on the solubility of curcumin. Ordinary one-way ANOVA and Tukey’s multiple comparison tests were performed to compare the formulations with each other. The * and ** indicate statistically significant differences at *p* < 0.05 and *p* < 0.01.

**Figure 2 molecules-27-06652-f002:**
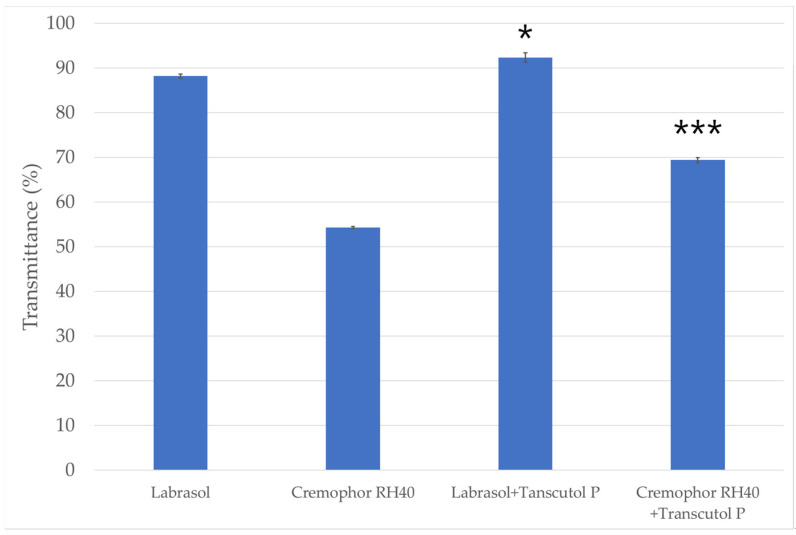
The emulsification efficiency of the surfactants and the mixture of surfactants and co-surfactant. Data are expressed as mean ± SD and *n* = 3. The usage of surfactants alone (Labrasol/Cremophor RH 40) was compared to the transmittance of the mixture containing co-surfactant (Transcutol P) with one-way ANOVA followed by Tukey’s multiple comparison test. Statistical significance is indicated *, *** at *p* < 0.005 and *p* < 0.001.

**Figure 3 molecules-27-06652-f003:**
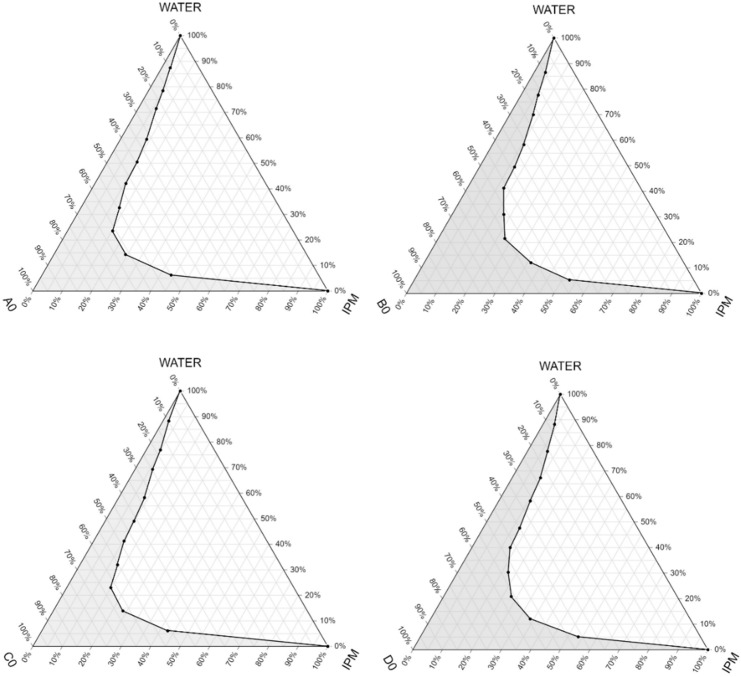
Pseudoternary phase diagrams of the compositions without the active ingredient. (Shaded areas represented the nano/microemulsion regions).

**Figure 4 molecules-27-06652-f004:**
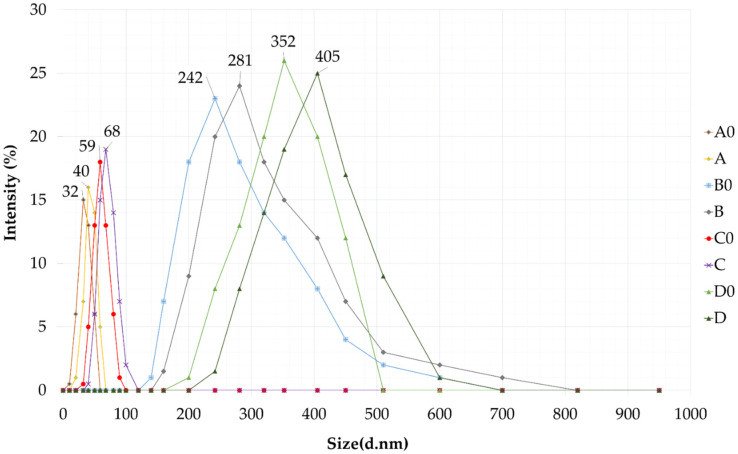
Droplet size of self-nano- and microemulsifying drug delivery systems (SNEDDS and SMEDDS) in water via dynamic light scattering (DLS) measurement.

**Figure 5 molecules-27-06652-f005:**
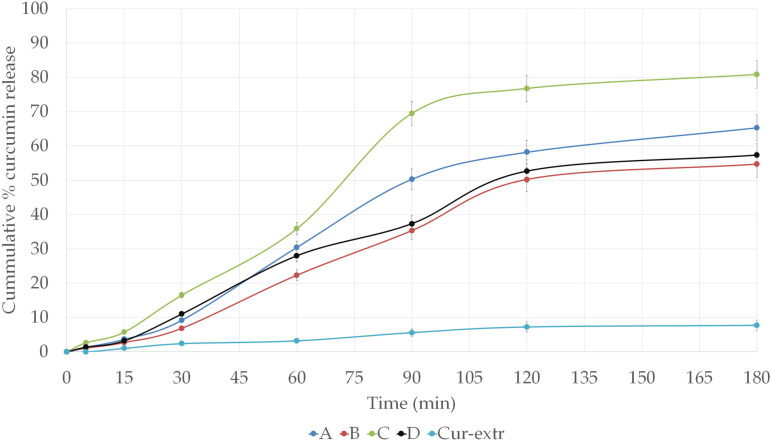
In vitro dissolution profiles of curcumin from the different SNEDDS and SMEDDS compositions and from curcumin extract loaded capsules in simulated intestinal fluid (SIF) without pancreatin (pH 6.8) at 37 °C.

**Figure 6 molecules-27-06652-f006:**
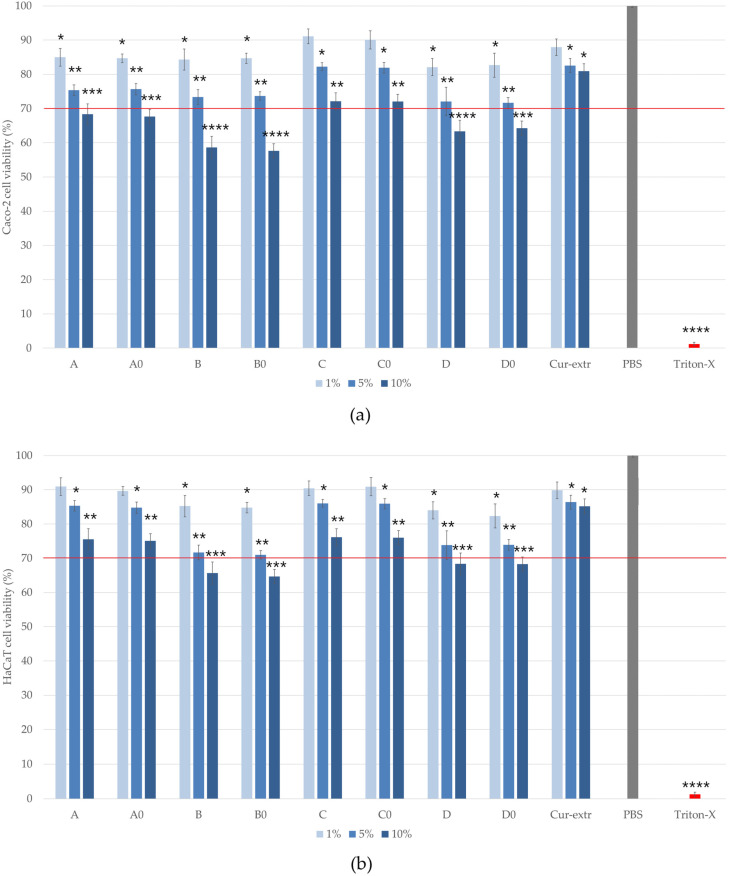
Cell viability test with the MTT assay on Caco-2 (**a**) and on HaCaT cells (**b**) after incubation with the formulations for 2 h. Cell viability is expressed as the percentage of negative control (PBS), which was treated only with PBS. The positive control was Triton X 100 (10% *w*/*v*), which had significantly lower cell viability results compared with the untreated control. Each data point represents the mean ± S.D. and *n* = 12. Ordinary one-way ANOVA with Dunnett’s multiple comparison test was performed to compare the different formulations with PBS. The *, **, ***, and **** indicate statistically significant differences at *p* < 0.05, *p* < 0.01, *p* < 0.001, and *p* < 0.0001.

**Figure 7 molecules-27-06652-f007:**
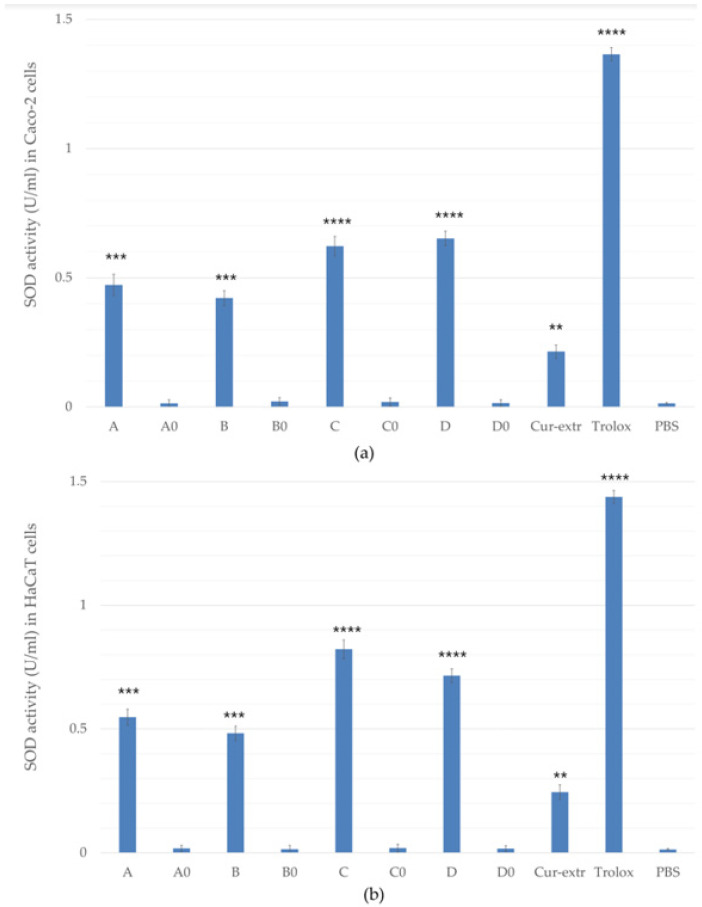
Effects of pretreatment with compositions on superoxide dismutase (SOD) enzyme activity in Caco-2 (**a**) and HaCaT cells (**b**). Cells treated with PBS were used as negative control. As positive control, Trolox (10.0 µM) dissolved in PBS was used. Data are expressed as the mean ± S.D. and *n* = 12. Ordinary one-way ANOVA with Dunnett’s multiple comparison test was performed to compare the different formulations with PBS. The **, ***, and **** indicate statistically significant differences at *p* < 0.01, *p* < 0.001, and *p* < 0.0001.

**Figure 8 molecules-27-06652-f008:**
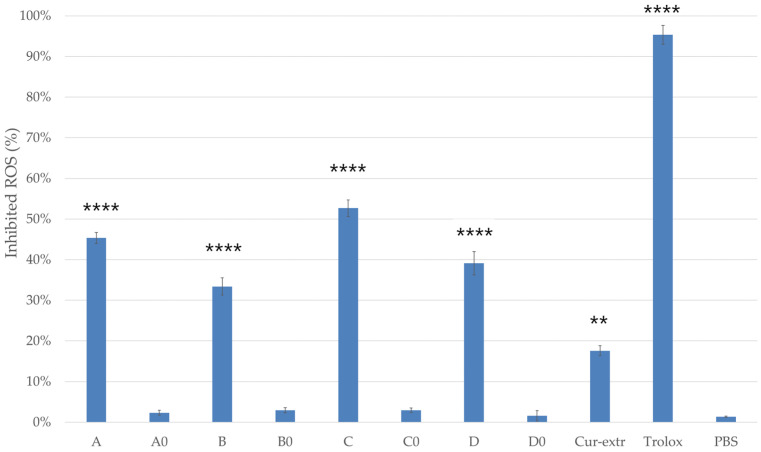
DPPH-scavenging activity of the compositions (5 *w/v*%). As positive control, Trolox (10.0 µM) dissolved in ethanol (96%) was used. As negative control, 2.0 mL of DPPH solution (0.06 mM) diluted with 1.0 mL absolute ethanol and PBS were applied. Data presented as mean ± SD (*n* = 6). Ordinary one-way ANOVA with Dunnett’s multiple comparison test was performed to compare the different formulations with PBS. The ** and **** indicate statistically significant differences at *p* < 0.01 and *p* < 0.0001.

**Figure 9 molecules-27-06652-f009:**
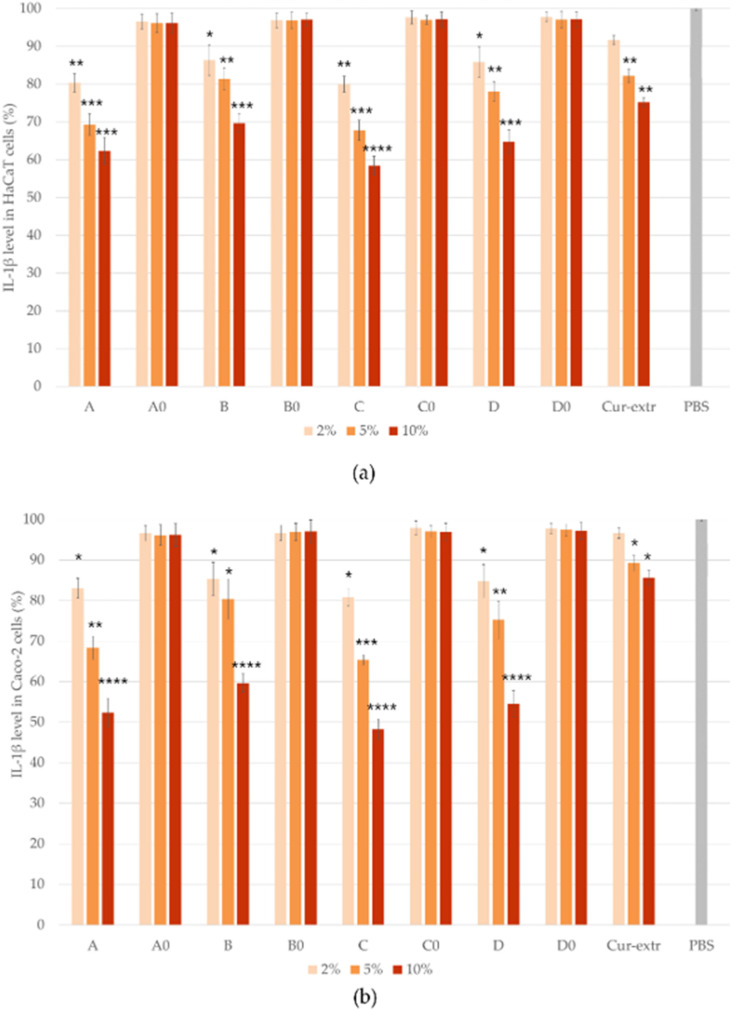
Results of human IL-1β ELISA tests on HaCaT (**a**) and Caco-2 (**b**) cells. Data represent the mean of six wells ± SD. Data are expressed as means ± SD, *n* = 6. Ordinary one-way ANOVA with Dunnett’s multiple comparison test was performed to compare the different formulations with PBS. The *, **, ***, and **** indicate statistically significant differences at *p* < 0.05, *p* < 0.01, *p* < 0.001, and *p* < 0.0001.

**Figure 10 molecules-27-06652-f010:**
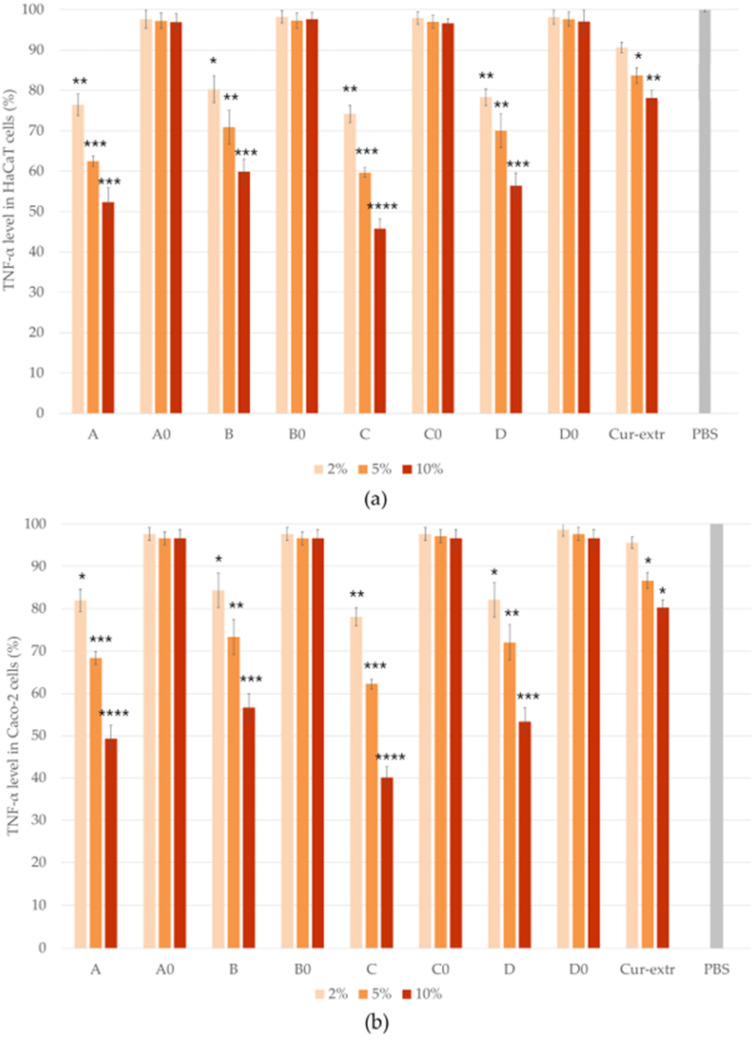
Results of human TNF- α ELISA tests on HaCaT (**a**) and Caco-2 (**b**) cells. Ordinary one-way ANOVA with Dunnett’s multiple comparison test was performed to compare the different formulations with PBS. The *, **, ***, and **** indicate statistically significant differences at *p* < 0.05, *p* < 0.01, *p* < 0.001, and *p* < 0.0001.

**Table 1 molecules-27-06652-t001:** Particle size, PDI, zeta potential and drug loading efficiency of the formulated SNEDDS/SMEDDS. Values are expressed as mean ± S.D., *n* = 3.

Composition	Droplet Size (nm)	PDI	Zeta Potential (mV)	Drug Loading Efficiency (%)
A0	32.29 ± 0.21	0.212 ± 0.016	−17.60 ± 0. 46	
A	40.81 ± 2.46	0.228 ± 0.024	−17.52 ± 0.24	98.39 ± 0.52
B0	242.12 ± 4.23	0.202 ± 0.008	−15.34 ± 0.32	
B	281.04 ± 6.02	0.204 ± 0.006	−15.12 ± 0.44	93.11 ± 0.25
C0	59.13 ± 3.65	0.179 ± 0.012	−18.12 ± 0.36	
C	68.45 ± 3.66	0.186 ± 0.014	−17.88 ± 0.38	99.12 ± 0.43
D0	352.02 ± 8.76	0.104 ± 0.006	−15.98 ± 0.28	
D	405.34 ± 12.84	0.112 ± 0.008	−16.02 ± 0.33	95.38 ± 0.33

**Table 2 molecules-27-06652-t002:** Results of the thermodynamic stability tests. The samples were subjected to six heating–cooling cycles between 4 °C and 45 °C and three freeze–thaw cycles at alternating temperatures of −21 °C and 25 °C. The formulations were centrifuged at 3500 rpm for 30 min.

Composition	Heating–Cooling Cycles	Centrifugation	Freeze–Thaw Cycles
A0	√	√	√
A	√	√	√
B0	√	√	√
B	√	√	√
C0	√	√	√
C	√	√	√
D0	√	√	√
D	√	√	√

**Table 3 molecules-27-06652-t003:** The compositions of the formulated self-nanoemulsifying systems.

Composition	Transcutol P	Cremophor RH 40	Labrasol	IPM	Curcumin
A0	37.5 g	37.5 g	-	15.0 g	-
A	37.5 g	37.5 g	-	15.0 g	10.0 g
B0	30.0 g	30.0 g	-	30.0 g	-
B	30.0 g	30.0 g	-	30.0 g	10.0 g
C0	37.5 g	-	37.5 g	15.0 g	-
C	37.5 g	-	37.5 g	15.0 g	10.0 g
D0	30.0 g	-	30.0 g	30.0 g	-
D	30.0 g	-	30.0 g	30.0 g	10.0 g

## Data Availability

The data that support the findings of this study are available from the corresponding author (feher.palma@pharm.unideb.hu) with the permission of the head of the department, upon reasonable request.

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
