# Peer review of "Enhanced Antioxidant and Anti-Inflammatory Effects of Self-Nano and Microemulsifying Drug Delivery Systems Containing Curcumin"

_molecules, 2022, doi:10.3390/molecules27196652_

Round 1

Reviewer 1 Report

The authors report compelling studies on improving the solubility and, thus, the bioavailability of curcumin - a known food component with antioxidative and anti-inflammatory activity. The authors have identified the formulation that could provide curcumin solubility relatively similar to the most effective formulation in oils. The established formulations were found to be non-toxic and deliver the impact from the curcumin. Overall, this is an interesting study that highlights the applicability of nano and microemulsifying delivery systems to non-polar bioactive compounds to improve their bioavailablity.

The manuscript is well written and provides sufficient experimental details that support the conclusions.The recommendation is to accept in present form.

Author Response

Dear Reviewer 1,

We would like to thank you for your kind words regarding our manuscript. We honestly hope that many researchers will be able to use our results for their own experiments. However, the manuscript is modified according to the requests of the other reviewers.

Sincerely,

Pálma Fehér

corresponding author

Reviewer 2 Report

The paper submitted by Jozsa et al. deals with the preparation and characterization of curcumin-loaded drug delivery systems based on self-nano and microemulsifying systems.

The manuscript is clear, well written and the conclusions are supported by the results. However, some corrections are needed before its publication:

1. L 52: its health benefits are mainly due...

2. L115: provide the value for the HLB and a reference

3. table 2: add in the caption the number of cycles

4. use the expression "heating-cooling cycles" within the entire manuscript

5. L217: revise "formulation C0 and C0.."

6. a non-cytotoxic sample has a cellular viability higher than 80%. for cellular viabilities between 70 and 80%, the sample is already moderate cytotoxic. photos of the cells can also be added.

7. several recent references must be added in the discussion section. some suggestions are: https://doi.org/10.3390/ijms22063075; https://doi.org/10.3390/polym12071450; https://doi.org/10.1016/j.ijbiomac.2019.12.247

Author Response

Dear Reviewer 2,

I hereby submit our modified manuscript “Enhanced Antioxidant and Anti-Inflammatory Effects of Self-Nano and Microemulsifying Drug Delivery Systems Containing Curcumin”, by Liza Józsa et al. Thank you for your constructive comments, according to them, the following changes were made during revision (corrections related to the Review 2 are marked with green in the manuscript):

  1. In line 54. the spelling mistake has been corrected (“its health benefits are mainly due to...”)
  2. In line 126. the HLB value of the surfactants has been provided. (“…Labrasol has a HLB value of 12, while Cremophor RH 40 has a value of 15.” [Safwat, S.; Ishak, R.A.H.; Hathout, R.M.; Mortada, N.D. Nanostructured lipid carriers loaded with simvastatin: effect of PEG/glycerides on characterization, stability, cellular uptake efficiency and in vitro cytotoxicity. Drug Dev. Ind. Pharm. 2017, 43, 1112–1125. , Patel, R.B.; Patel, M.R.; Bhatt, K.K.; Patel, B.G. Formulation consideration and characterization of microemulsion drug delivery system for transnasal administration of carbamazepine. Bull. Fac. Pharmacy, Cairo Univ. 2013, 51, 243–253.]
  3. In the caption of Table 2 number of cycles has been added: “Table 2. Results of the thermodynamic stability tests. The samples were subjected to six heating-cooling cycles between 4 and 45 °C and three freeze-thaw cycles at alternating temperatures of -21 and 25 °C. The formulations were centrifuged at 3500 rpm for 30 min.
  4. The expression "heating-cooling cycles" was used the entire manuscript, as it was suggested.
  5. In line 249. "formulation C0 and C0..." has been corrected to “formulation C0 and C…”
  6. According to the ISO 10993:5 recommendations, if the cell viability value is over 70% (examined by MTT test), the preparation can be called biocompatible and non-cytotoxic. [International Organization for Standardization Geneva Switzerland ISO/EN 10993-5 Biol. Eval. Med. devices - Part 5 Tests Cytotox. Vitr. methods; 3rd ed.; 2009.] Unfortunately, MTT assay is not suitable for cellular image taking.
  7. The suggested references has been added in the discussion section.

We hope that our improved manuscript is suitable for publication.

Waiting for your kind response.

Sincerely,

Pálma Fehér

corresponding author

Reviewer 3 Report

The study by Józsa et al., titled “Improved antioxidant and anti-inflammatory effect of Self-Nano and Microemulsifying Drug Delivery Systems containing curcumin”. is well written. However, to be considered for publication in the Molecules Journal, the authors must revise the manuscript in accordance with the suggestions made in the comments below.

 1.    The authors should consider changing the title as mentioned below. Use enhanced, which is more appropriate, rather than improved.

Enhanced Antioxidant and Anti-Inflammatory Effects of Self-Nano and Microemulsifying Drug Delivery Systems Containing Curcumin”

2.    Abstract: Rewrite the abstract with care. It is not a true reflection of the overall research findings from this study. The authors mostly concentrated on and presented methods. I couldn't find much information about the results. The abstract's final four lines similarly mostly discussed methods. Moreover, I was unable to locate a conclusion segment in the abstract. In addition, include future direction of their finding at the end.

3.    Introduction: Reference No. 1 is outdated. The authors considered that it would be best to incorporate a more recent reference in lines 41–48 https://doi.org/10.3389/fphar.2022.820806

 4.    I was unable to find a specific research gap outlining why self-nano and microemulsifying drug delivery systems were chosen and what their merits to curcumin.  Similarly, many of other drug delivery methods were used for curcumin in past studies to improve its solubility and bioavailability, how well does your research overcome the drawbacks of those earlier studies as documented in the literature? It is essential to provide all of these details in the introduction.

 5.    The section title of 2.6, 2.7 and 2.8 needs to be modified. Remove “Results of the” in all the three titles. Since, this is already in the results section.

 6.    In my opinion, section 2.7 supposed to be In vitro dissolution study, then followed by 2.8, 2.9 and 3.10 to be in-vitro results.

 7.    The results section is written effectively overall. The discussion part, however, requires significant improvements. It was difficult for me to focus on it. I suggest author takes out the repetition words in the first two paragraphs with introduction, and use clear, succinct words throughout the discussion to make it easier for readers to understand what the authors are really talking about it.

 8.    Conclusion: The author should enhance the work's novelty (in the conclusion section). The findings/insights should be used to support this section. Finally, it will give a clear understanding of the study. Future viewpoints need to be covered in the conclusion as well. The value of the research should be highlighted by the author.

Author Response

Dear Reviewer 3,

I hereby submit our modified manuscript “Enhanced Antioxidant and Anti-Inflammatory Effects of Self-Nano and Microemulsifying Drug Delivery Systems Containing Curcumin”, by Liza Józsa et al. Thank you for your constructive comments, according to them, the following changes were made during revision (corrections related to the Review 3 are marked with yellow in the manuscript):

  1. The title has been changed as mentioned: “EnhancedAntioxidant and Anti-Inflammatory Effects of Self-Nano and Microemulsifying Drug Delivery Systems Containing Curcumin”
  2. The abstract has been rewritten as it was suggested. We put more emphasis on the description of the results. A conclusion segment and future directions were added at the end of the abstract.
  3. The suggested reference (https://doi.org/10.3389/fphar.2022.820806) has been added to the beginning of the introduction part.
  4. The reasons why we used self-nano and microemulsifying drug delivery systems during the formulation were described with more care in the introduction. (lines 83-90)
  5. The section title of 2.6, 2.7 and 2.8 has been modified as it was suggested.
  6. According to the suggestion that section 2.7 should be the “In vitro dissolution study”, it has been changed in the results section as well as in the methods section of the manuscript.
  7. The discussion part of the manuscript has been rewritten in order to be more clear and easier for readers to understand. The repetition words in the first two paragraphs were deleted.
  8. The conclusion part has been fully rewritten. The novelty of the work has been enhanced, the value of the research has been highlighted and future viewpoints have been added as it was suggested.

We hope that our improved manuscript is suitable for publication.

Waiting for your kind response.

Sincerely,

Pálma Fehér

corresponding author

Round 2

Reviewer 3 Report

The author responded to all of my inquiries and revised the manuscript content accordingly. As a result, I suggest that it be taken into consideration for publication in Molecules Journal in its current form.